# Prognostic Significance of Peritumoral and Intratumoral Lymphatic Vessels Density in Clinically Node-Negative (cN0) Oral Squamous Cell Carcinoma: A Preliminary Report

**DOI:** 10.3390/medicina61091712

**Published:** 2025-09-19

**Authors:** Boris Kos, Petar Suton, Danko Müller, Vid Mirošević, Matija Mamić, Ivica Lukšić

**Affiliations:** 1Department of Maxillofacial Surgery, University Hospital Dubrava, School of Medicine, University of Zagreb, Avenue Gojko Šušak 6, 10000 Zagreb, Croatia; bkos@kbd.hr (B.K.); matija.mamic90@gmail.com (M.M.); 2Division of Oncology and Radiotherapy, University Hospital Dubrava, Avenue Gojko Šušak 6, 10000 Zagreb, Croatia; petarsuton@yahoo.com; 3Department of Pathology, University Hospital Dubrava, School of Medicine, University of Zagreb, Avenue Gojko Šušak 6, 10000 Zagreb, Croatia; predstojnik.pat@kbd.hr; 4Department of Pathophysiology, School of Medicine, University of Zagreb, Mijo Kišpatić Street 12, 10000 Zagreb, Croatia; vid.mirosevic@gmail.com

**Keywords:** oral squamous cell carcinoma, lymphovascular density, occult lymph node metastasis, elective neck dissection, podoplanin

## Abstract

*Background and Objectives*: Oral squamous cell carcinoma (OSCC) is characterized by a high propensity for cervical lymph node metastasis, which remains a strong predictor of patient outcome. Despite advances in management, the prognosis for OSCC has not significantly improved, and the identification of reliable predictors for occult lymph node metastasis (OLNM) in clinically node-negative (cN0) patients is crucial for optimizing treatment strategies. Lymphovascular density (LVD) immunohistochemically assessed by podoplanin (D2-40) has been proposed as a potential biomarker for regional metastasis, but its prognostic value remains controversial. This study aimed to evaluate the prognostic significance of intratumoral (ILVD) and peritumoral lymphovascular density (PLVD) for OLNM in OSCC. *Materials and Methods*: A retrospective analysis was conducted on 43 cN0 patients with primary OSCC who underwent surgical resection and elective neck dissection (END) at a tertiary care cancer center. LVD was assessed by immunohistochemical staining for podoplanin (D2-40) in both intratumoral and peritumoral regions. Clinicopathological data were collected and statistically analyzed. *Results*: In observed cohort peritumoral LVD was significantly higher than intratumoral LVD. PLVD was also significantly higher in early-stage tumors (pT1/pT2) compared to advanced stages (pT3/pT4). Higher ILVD was significantly associated with the presence of OLNM. Neither ILVD nor PLVD demonstrated a statistically significant influence on overall survival, although a trend toward poorer outcomes was observed in patients with higher ILVD. *Conclusions*: ILVD was significantly associated with occult nodal metastasis, whereas PLVD was not. However, neither LVD parameter independently predicted overall survival. Results suggest that ILVD may serve as a useful marker for identifying cN0 OSCC patients at higher risk for occult metastasis.

## 1. Introduction

Oral squamous cell carcinoma (OSCC) is the most common malignancy of the oral cavity [1,2]. It is the eight most common cause of death among all malignant tumors [2,3,4]. Despite ongoing efforts, the prognosis of OSCC has remained largely unchanged for decades [1,3]. The overall 5-year survival rate for OSCC in the United States is estimated to be between 50% and 60% across all stages of the disease. However, it is important to note that the majority of OSCC cases are diagnosed at advanced stages [4,5].

OSCC is characterized by a high rate of recurrence and demonstrates a pronounced tendency to metastasize, primarily through the lymphatic system [6]. Cervical lymph node metastasis remains strong predictor of outcome in OSCC [7,8,9].

A clinically node-negative neck (cN0) is defined as the absence of detectable lymph node metastases based on thorough preoperative assessment, including both clinical examination and radiological imaging modalities. Occult lymph node metastasis (OLNM) refers to metastatic deposits that are present but remain undetected by preoperative evaluation. This diagnostic challenge is underscored by the fact that approximately 20% to 30% of patients classified as cN0 are subsequently found to harbor OLNM upon pathological examination. When considering management strategies for the cN0 neck, tree options exist: elective neck dissection (END), sentinel lymph node biopsy (SLNB) and watchful waiting approach. Advocates of the watchful waiting strategy argue that this method spares approximately 70% to 80% of patients from undergoing unnecessary surgical intervention and its associated morbidity [6,10,11,12,13,14,15]. On the other hand, some authors reported higher survival rates in subgroup of patients treated with END compared to patients treated with neck observation and curative treatment during follow-up [16]. SLNB in OSCC with a clinically node-negative neck is a minimally invasive procedure that allows targeted identification and analysis of the first lymph node likely to harbor metastasis, showing high sensitivity and negative predictive value while reducing morbidity compared to END. However, it has limitations including technical complexity, the need for specialized equipment and personnel, potential difficulties in certain tumor locations, and often requires a second surgical procedure if metastasis is detected [15,17,18].

While multiple predictive factors for OLNM have been investigated, only depth of invasion (DOI) has constantly emerged as an independent predictive factor [9,19,20,21,22,23].

Consequently, for the long time considerable scientific efforts have been dedicated to the identification and validation of potential biomarkers capable of reliably detecting OLNM in patients with OSCC [9,24,25]. The discovery of such biomarkers would facilitate improved risk stratification and more personalized management of clinically node-negative necks, thereby optimize patient outcomes while minimize unnecessary treatment-related morbidity.

Tumor lymphangiogenesis represents a critical step in the development of lymph node metastasis, as it facilitates the dissemination of malignant cells through newly formed lymphatic channels. This process encompasses the formation of both intratumoral and peritumoral lymphatic vessels [9,24,25].

Lymphovascular density (LVD) has been proposed as a potential biomarker for regional metastasis in OSCC. However, the current literature shows conflicting evidence regarding whether intratumoral lymphovascular density (ILVD), peritumoral lymphovascular density (PLVD), or both play significant roles in metastasis or serve as prognostic indicators [9,26,27,28].

Podoplanin is a transmembrane glycoprotein expressed on lymphatic endothelial cells. Current literature highlights its dual role in both the development of the lymphatic vascular system and tumorigenesis [29,30,31,32,33,34]. Early research faced challenges in distinguishing lymphatic from blood vessels until the discovery of the anti-podoplanin antibody (D2-40) [35]. Since then, D2-40 has emerged as an established immunohistochemical marker for the research of lymphatic vessels [28,29,30,36,37,38,39,40,41].

This study seeks to assess the prognostic significance of PLVD and ILVD in predicting OLNM in OSCC. To achieve this, we utilize the D2-40 antibody, an established immunohistochemical marker for lymphatic endothelium, to accurately quantify lymphangiogenesis and explore its association with metastatic risk.

## 2. Materials and Methods

Forty-three patients diagnosed with OSCC who were treated consecutively with primary surgery at a tertiary care cancer center were included in this retrospective study. Clinical and pathological data for these patients were collected from the institutional database, covering the period from January 1, 2010, to January 1, 2015. The inclusion criteria were as follows: (a) T1–T4 primary OSCC; (b) no previous history of Head and Neck Squamous Cell Carcinoma (HNSCC); (c) surgical resection of the primary tumor with concomitant END; (d) absence of neoadjuvant treatment; (e) no radiological or clinical evidence of LNM; and (f) reliable follow-up. Exclusion criteria were as follows: (a) OSCC patients with a clinically node-positive neck (cN+); (b) previous surgical or chemo/radiotherapy of HNSCC; (c) unreliable follow-up; and (d) incomplete data in hospital medical records. Preoperatively, all patients underwent clinical and radiological evaluation using either multi-slice computed tomography (MSCT) or magnetic resonance imaging (MRI). Standard preoperative protocol included contrast-enhanced imaging of the head, neck and thorax. All cases were presented preoperatively at the institutional multidisciplinary tumor board. The indication for END has been routinely established for locally advanced primary tumors (cT3, cT4). The decision to perform END in cT2 tumors has been made considering tumor sublocation inside the oral cavity, histological grade, and the necessity for microvascular reconstruction. The standard of care for END in oral cancer includes dissection of levels I–III/IV, depending on the tumor prognosticators. Postoperatively, the neck specimens were divided into neck levels and sent for histopathological analysis. During follow-up, each patient was evaluated every 2–3 months for the first two years, every 4-6 months for the subsequent four years, and annually thereafter. Each follow-up appointment included a physical examination. Ultrasound of the region was performed 1–2 times per year, and the imaging (MSCT or MRI) of the head, neck and thorax was conducted annually.

### 2.1. Histological and Immunohistological Analysis

Postoperative histopathological analysis was routinely carried out on hematoxylin and eosin (HE)-stained slides by an experienced pathologist specializing in head and neck pathology (D.M.). The HE staining technique employed for routine histological evaluation involves sequential application of hematoxylin, which stains cell nuclei blue purple, followed by eosin, which stains cytoplasmic and extracellular proteins in varying shades of pink. The tissue sections were prepared from formalin-fixed, paraffin-embedded specimens, cut at a standard thickness of 4–5 μm. For the purposes of this study, several key histopathological parameters were documented. These included tumor T staging based on the updated 8th edition of the American Joint Committee on Cancer (AJCC) TNM classification system. Additional parameters recorded were the presence of perineural invasion and perivascular invasion, tumor sublocation within the oral cavity, maximum tumor diameter, and the presence or absence of extracapsular extension (ENE) of nodal metastases. Immunohistochemical analysis was performed on four-micrometer-thick sections of paraffin-embedded primary tumor tissue using the anti-podoplanin antibody D2-40 (Agilent, M3619). The D2-40 antibody binds specifically to podoplanin, a mucin-type transmembrane glycoprotein expressed on lymphatic endothelium but not on blood vessels. The immunostaining procedure involved antigen retrieval, typically by heat-induced epitope retrieval using citrate buffer, followed by incubation with the primary antibody and detection via a polymer-based secondary antibody system with chromogenic visualization using diaminobenzidine (DAB). This protocol yields distinct brown immunoreactive lymphatic vessels suitable for quantitative analysis. LVD was assessed by identifying lymphatic vessel hotspots both within the tumor mass (intratumoral) and in the peritumoral stroma, defined as tissue located within 0.5 mm of the tumor margin. This evaluation followed the method described by Yuan et al. [42], wherein the areas with the highest concentration of lymphatic vessels are systematically counted (Figure 1 and Figure 2). The density was quantified over an area of 3 mm^2^, corresponding to 12 high-power fields at 400× magnification, using an Olympus BX46 microscope (Olympus Optical Co., Tokyo, Japan). This semiquantitative assessment by a single experienced pathologist ensured consistency and minimized interobserver variability.

### 2.2. Statistical Analysis

Statistical analyses for this study were performed using GraphPad Prism software, version 8.4.3 (GraphPad Software, San Diego, CA, USA).

Descriptive statistical analysis was performed on a study cohort consisting of 43 patients. Age data were summarized by calculating the mean and median values. Gender distribution and tumor localization were presented as counts and percentages. Tumor sites included sublingual region, tongue, retromolar trigone, and mandibular gingiva. Tumor staging was determined according to the AJCC 8th edition criteria for oral cancer [43], with the pathological T classification (pT1 to pT4b) recorded for each patient. Patients were further subcategorized according to the Union for International Cancer Control (UICC)/AJCC 8th edition overall stages of the disease (I–IV). The presence of perineural invasion (PNI) and lymphovascular invasion (LVI) was documented as dichotomous variables. Tumor diameter was measured and expressed as mean and median values. Surgical margin status was classified into clear margins (R0, >5 mm) and close margins (R1, 2–5 mm), with corresponding frequencies reported. Pathological evaluation of lymph node status included recording the presence or absence of lymph node metastases and ENE in positive cases. Treatment failure was defined by occurrence of local recurrence, regional recurrence, or distant metastasis during follow-up. The frequencies of these failure types were reported as percentages of the total cohort and subcategories. The development of second primary tumors during follow-up was similarly documented. Disease-free survival (DFS) and overall survival (OS) times were calculated in months, with both mean values, standard deviations, median values, and interquartile ranges (IQR) reported. ILVD and PLVD were quantified as the number of lymphatic vessels per 3 mm^2^ and summarized by means and medians to describe their distribution within the cohort.

For the detailed statistical analysis, the initial step consisted of evaluating the distribution of continuous variables to determine whether they adhered to a normal distribution. This was systematically assessed using the Shapiro–Wilk test, which was applied to each continuous variable included in the analysis.; this was accomplished using the Shapiro–Wilk test. Data points that deviated significantly from the overall dataset were identified as potential outliers through application of the Grubbs test, allowing for the detection and consideration of anomalous values that could potentially bias results. Categorical variables were summarized as counts and percentages to provide a clear overview of the distribution of categorical data within the study population. Comparisons between categorical groups were made using either Fisher’s exact test, which is appropriate for small sample sizes and provides exact *p*-values, or the chi-square (χ^2^) test, applicable when sample sizes were sufficiently large, ensuring an accurate assessment of associations between categorical variables. To compare the means of continuous variables between two independent groups, a two-tailed unpaired Student’s *t*-test was employed. For stratifying LVD into distinct subgroups reflecting lower and higher density, receiver operating characteristic (ROC) curve analysis was utilized separately for tumor-associated LVD and stromal LVD. Within the analyzed cohort, the discriminatory capacity of LVD was evaluated concerning the patients’ survival status, categorized as “alive” or “deceased,” during the follow-up period. The optimal cutoff values differentiating low versus high LVD groups were determined using the Youden index, which maximizes the sum of sensitivity and specificity to identify the point that best separates outcomes. The baseline for survival analyses was defined as the date of surgical treatment, from which the time to occurrence of predefined events of interest was measured. Specifically, the study focused on three-year and five-year OS as key endpoints. Time-to-event data were analyzed employing the Kaplan–Meier method, which estimates survival probabilities over time while accounting for censored observations such as patients lost to follow-up. Differences between survival curves of stratified groups were statistically compared using the log-rank test.

## 3. Results

A total of 43 patients were included in this study. The mean age was 61.3 years, with a median age of 59 years. Most patients were male, comprising 86.0% (*n* = 37) of the cohort. Tumor localization was distributed as follows: 23.2% (*n* = 10) had sublingual cancer, 30.2% (*n* = 13) had tongue cancer, 25.6% (*n* = 11) had retromolar trigone cancer, and 21.0% (*n* = 9) had mandibular gingiva cancer. On definitive pathological analysis, 21.0% (*n* = 9) of patients had pT1 tumors, 48.8% (*n* = 21) had pT2, 9.3% (*n* = 4) had pT3, 18.6% (*n* = 8) had pT4a, and 2.3% (*n* = 1) had pT4b tumors. Regarding the UICC overall stage, 13.9% of patients (*n* = 6) were classified as stage I, 30.2% (*n* = 13) as stage II, 21.0% (*n* = 9) as stage III, and 34.9% (*n* = 15) as stage IV. PNI and LVI were present in 55.8% (*n* = 24) of patients. The mean tumor diameter was 3.04 cm, with a median value of 3.0 cm. Considering surgical margins, 81.4% (*n* = 35) of patients had clear margins R0 (greater than 5 mm), while the remaining patients had close margins R1 (between 2 and 5 mm). Lymph node metastases were identified on pathological evaluation in 34.9% (*n* = 15) of patients, and 60.0% (*n* = 9) of these exhibited extranodal extension. Treatment failure was defined as local or regional tumor recurrence or distant metastasis during follow-up. A total of 25.6% (*n* = 11) of patients experienced treatment failure during follow-up: 72.7% (*n* = 8) had local failure, 9.1% (*n* = 1) had regional failure, and 18.2% (*n* = 2) had distant failure. Additionally, 18.6% (*n* = 8) of patients developed a second primary tumor. The mean DFS was 29.2 ± 23.2 months. The median DFS was 29 months, with an interquartile range of 42.5 months. The mean overall survival OS was 32.8 ± 22.3 months. The median OS was 36 months, with an IQR of 35.5 months. The mean ILVD was 15.8 vessels per 3 mm^2^, with a median of 14 vessels per 3 mm^2^. The mean peritumoral lymphatic vessel density was 28.8 vessels per 3 mm^2^, with a median of 30 vessels per 3 mm^2^.

Analyses revealed a statistically significant difference between PLVD and ILVD, with higher values observed in the peritumoral regions compared to the intratumoral regions (*p* < 0.05; Figure 3).

There was no significant difference in ILVD in relation to tumor T stage, as defined by the AJCC 8th edition staging system for oral cancer (pT1 and pT2 vs. pT3 and pT4a/b; *p* > 0.05). In contrast, PLVD was significantly higher in smaller tumors (pT1 and pT2) compared to locally advanced tumors (pT3 and pT4a/b) (*p* < 0.05; Figure 4).

Both ILVD and PLVD were evaluated for correlation with OLNM in the final pathological report. Higher ILVD was significantly associated with the presence of occult metastasis (*p* < 0.05), whereas no significant correlation was observed between PLVD and occult metastasis (*p* > 0.05; Figure 5).

Patients were stratified into two groups according to overall AJCC/UICC 8th edition staging manual: stage I and II (*n* = 19) and stage III and IV (*n* = 24). There was no significant difference in either ILVD or PLVD between patients with low-stage (stage I–II) and high-stage (stage III–IV) disease (*p* > 0.05 for both comparisons; Figure 6).

Patients were subsequently divided into two groups according to ILVD, based on an optimal cutoff value of 8.5 vessels per 3 mm^2^ as determined by ROC analysis using the Youden index: ILVD < 8.5 and ILVD ≥ 8.5. Survival analysis performed using the log-rank (Mantel–Cox) test, demonstrated no statistically significant difference between the groups (*p* > 0.05), although a trend toward poorer outcomes was noted in patients with higher ILVD (Figure 7). Similarly, patients were stratified according to peritumoral LVD using a cutoff value of 19 vessels per 3 mm^2^, also determined by ROC analysis and the Youden index. Survival analysis revealed no significant difference between patients with PLVD < 19 and those with PLVD ≥ 19 (*p* > 0.05, Figure 8).

## 4. Discussion

The main scope of this study is to determine the influence of intratumoral and peritumoral lymphatic vessel density (ILVD and PLVD) as prognostic factors for OLNM in oral squamous cell carcinoma OSCC.

The propensity for epithelial malignancies to metastasize via lymphatic vessels arises from their structural characteristics. Thinner walls composed of a single endothelial layer and discontinuous basement membrane, resulting in greater permeability compared to blood vessels [30,44]. Lymph vessels are continuously created and destroyed in the process of cancerogenesis [30,45,46]. Contemporary understanding posits that these vessels function not merely as passive conduits for metastasis but actively participate in complex molecular interplay [47]. Consequently, increased lymphatic vessel density may facilitate more efficient cancer dissemination [30,48,49].

In our study, we observed significantly higher PLVD compared to ILVD. This contrasts with findings by Mermond et al. [9], who reported higher ILVD counts. The discrepancy may be attributable to their use of PROX1—a nuclear marker—which differs from our detection methodology. Another important distinction is that their study included both oral and oropharyngeal cancers, whereas our research focused exclusively on oral cancer. There may be biological differences between these tumor sites that influence the interaction between lymphatic vessels and tumor cells, particularly since human papillomavirus (HPV) status was not mentioned in their analysis. Chen et al. [2] also reported significantly higher PLVD in peritumoral region. The discrepancy observed in different studies may be attributable to differences in methodology. On the other hand, some authors reported no significant difference between PLVD and ILVD [36]. This discrepancy could also be attributed to the omission of certain tumor microenvironmental factors in studies, which may act as potential confounders when investigating lymphatic vessel density.

In our study, PLVD was significantly higher in early-stage tumors (pT1 and pT2) compared to locally advanced tumors (pT3 and pT4). No significant difference was observed for ILVD across tumor stages. This aligns with research by Mafra et al. [50], which found no correlation between ILVD and pT stage, though they reported higher ILVD in larger tumors (pT3 and pT4). Conversely, Franchi et al. [51] noted no significant association between pT stage and either PLVD or ILVD, although their results approached statistical significance (*p* = 0.07 and *p* = 0.06, respectively). Jardim et al. [52] had a trend towards statistical significance between the ILVD and pT (*p* = 0.06). Our findings suggest that elevated PLVD in smaller, potentially less aggressive tumors may reflect a protective mechanism, consistent with hypotheses that high LVD could facilitate anti-tumor immunity through immune cell recruitment. [53]

In our study, there was no significant correlation between overall disease stage, as defined by the UICC/AJCC 8th edition staging manual, and either ILVD or PLVD. Although there was a trend toward higher ILVD and PLVD in patients with advanced-stage disease (stage III and IV) compared to those with early-stage disease (stage I and II), the differences were not statistically significant. Similar findings have been reported by other authors, who also failed to demonstrate a significant correlation between lymphatic vessel density and disease stage [2]. Conversely, Kyzas et al. [27] observed higher ILVD in patients with advanced-stage disease. It is possible that our results could reach statistical significance in a larger cohort, although this remains speculative.

In our current study higher ILVD was significantly associated with the presence of OLNM. Some studies have correlated PLVD with regional metastasis in head and neck squamous cell carcinoma [9,26,27,51]. Conversely, other studies have demonstrated a positive correlation between ILVD and regional metastasis in head and neck cancer, consistent with our findings [27,28,54,55,56]. Interestingly, some studies have identified both ILVD and PLVD as significant prognostic marker for lymph node metastasis [2,27]. Such disparities in findings may be attributed to methodological inconsistencies, subjectivity in immunohistochemical analysis, and heterogeneity in tumor subsites within the head and neck region. It is described in the current literature that there are distinct differences between intratumoral and peritumoral lymphatic vessels in both their morphology and biology [9,24,25]. The role of pre-existing lymphatic vessels in the peritumoral stroma versus newly formed intratumoral lymphatic vessels in cancer dissemination remains controversial [27,54,57]. Some authors suggest that intratumoral lymphatic vessels are non-functional and that metastatic spread therefore occurs via peritumoral stromal vessels [58]. In other non-head and neck malignancies, research suggests that cancer cells metastasize exclusively via peritumoral lymphatic vessels [59]. This discrepancy may be explained by differences in cancer biology between various anatomical subsites [27].

In our study, there was no significant correlation LVD and survival outcome, although higher ILVD showed a trend toward poorer outcomes. Some studies have also failed to demonstrate a correlation between intratumoral or peritumoral lymphatic vessel density and survival outcomes [51]. Contrary, other studies have reported a significant association between ILVD and reduced survival [2,52,60,61].

Controversy persists in this field, with some authors correlating higher PLVD with worse outcomes [51], while others associate higher PLVD with more favorable prognoses. The latter group explains this phenomenon by proposing that lymphatic vessels facilitate immune cell recruitment [61]. A larger patient cohort in future studies may reveal a significant correlation in our population as well.

### 4.1. Future Research Directions

This research contributes valuable insights to the complex and evolving field of LVD as a prognostic factor in OSCC. Notably, it focuses on a specific patient population with clinically node-negative status, who represent a critical subgroup of high interest in current head and neck oncology research due to the challenge of OLNM prediction. While the potential of LVD to serve as a reliable prognostic biomarker for OLNM is promising, its clinical utility remains to be conclusively established. Prospective studies involving larger patient cohorts are essential to validate these preliminary findings and clarify the prognostic value of LVD in this context. Additionally, future investigations should explore the feasibility of accurately assessing LVD from routine preoperative biopsy specimens, which could enhance clinical decision-making without requiring extensive surgical sampling. This includes research into technical advancements and standardization of immunohistochemical techniques, as well as the identification of complementary molecular markers that may refine LVD determination and improve predictive accuracy. Such efforts will be crucial to translate LVD assessment into a practical tool for personalized management in OSCC patients.

### 4.2. Limitations of the Study

Although our study contributes to existing knowledge on the controversial topic of LVD in OSCC, several limitations should be acknowledged. First, the retrospective design—while consistent with other studies in this field—introduces inherent biases; prospective validation in a larger cohort would provide more robust and clear conclusions. Second, the relatively small cohort size may limit statistical power; expanding the sample size could yield more significant results and enhance understanding of LVD’s prognostic role. Third, reliance on a single immunohistochemical method may affect generalizability, as significant variability exists in LVD assessment due to divergent staining protocols and quantification techniques across studies. Standardization of these methodologies would improve result consistency and interpretability.

## 5. Conclusions

In our cN0 OSCC cohort, PLVD was significantly higher than ILVD. PLVD was also significantly higher in early-stage tumors (pT1/pT2) compared to advanced stages (pT3/pT4). ILVD showed a significant association with OLNM. No significant influence of LVD on OS was observed, although ILVD demonstrated a trend toward poorer outcomes.

## Figures and Tables

**Figure 1 medicina-61-01712-f001:**
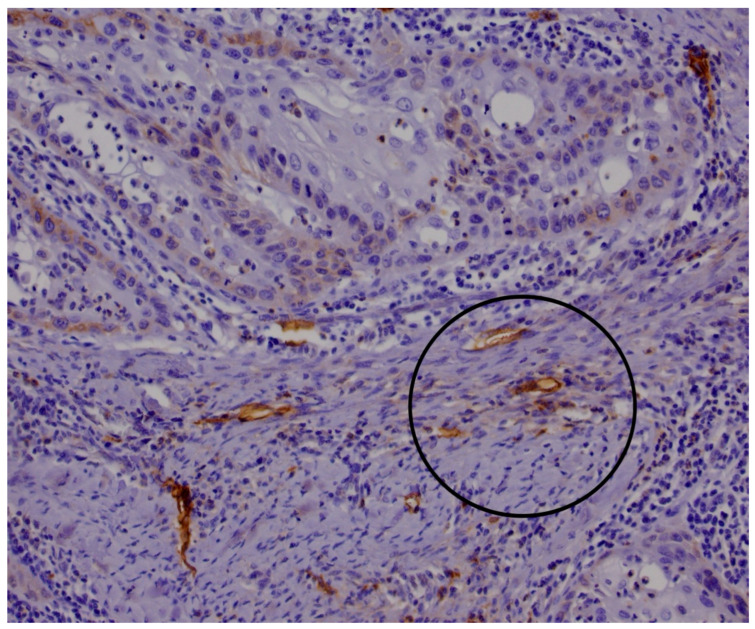
D2-40 immunohistochemical staining demonstrating positive reactions in lymphatic vessel endothelium within the tumor mass. Black circles highlight representative areas of positive reactivity used for tumor lymphatic evaluation, consistent with established methodologies.

**Figure 2 medicina-61-01712-f002:**
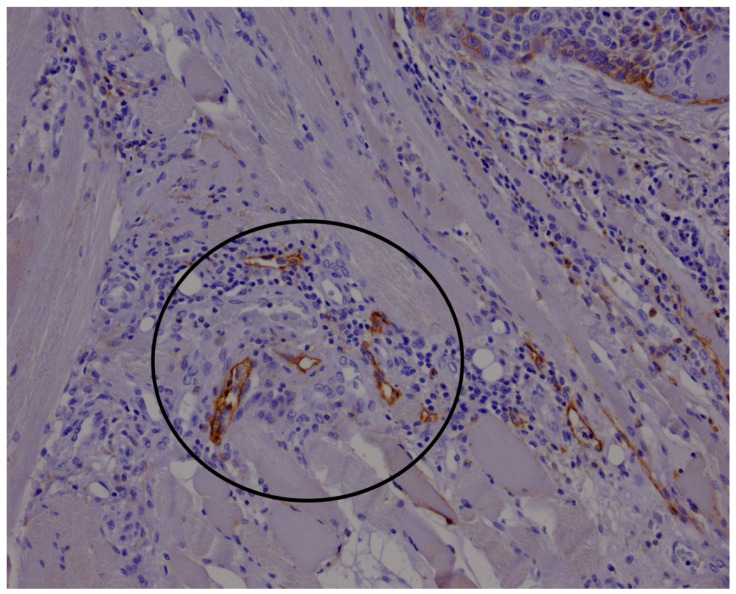
D2-40 immunohistochemical staining demonstrating positive reactivity in lymphatic vessel endothelium within peritumoral stroma (≤5 mm from tumor border). Black circles highlight representative areas used for lymphatic evaluation, consistent with established methodologies.

**Figure 3 medicina-61-01712-f003:**
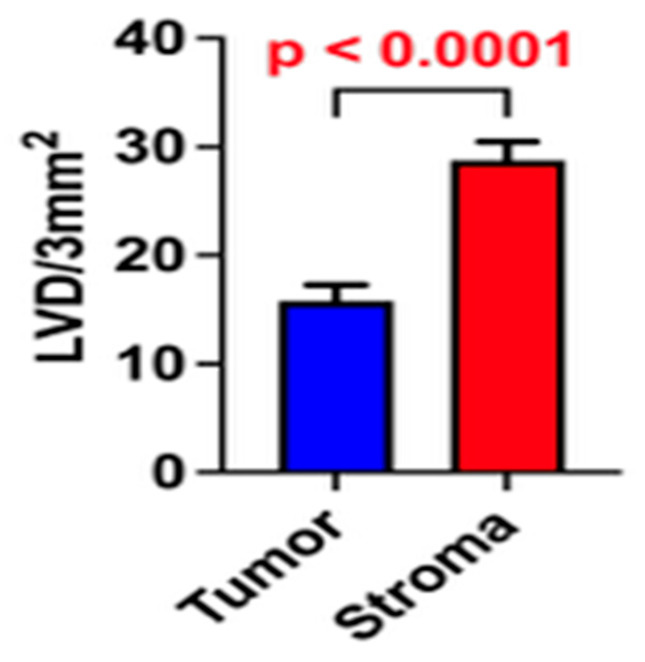
Comparison of lymphatic vessel density in tumor tissue and surrounding stroma. Data are means ± SEM. *p* < 0.05 was considered significant. LVD—lymphatic vessel density.

**Figure 4 medicina-61-01712-f004:**
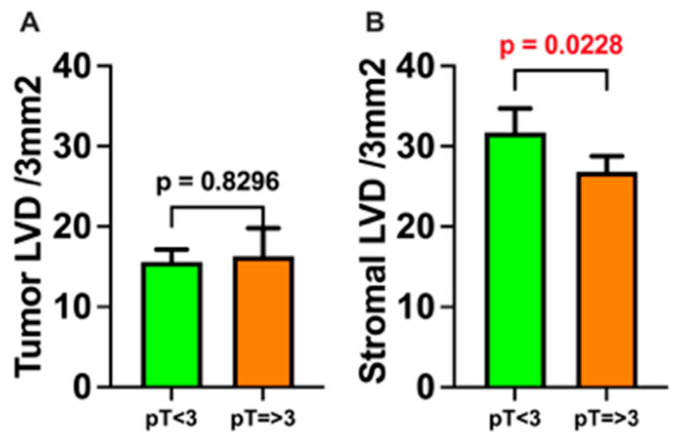
Intratumoral and stromal lymphatic vessel density (LVD) in relation to primary tumor size. Tissue samples were stratified into two groups according to pathological tumor size classification: pT < 3 (*n* = 30) and pT ≥ 3 (*n* = 13). (**A**) Lymphatic vessel density within tumor tissue (ILVD) based on pT group. (**B**) Lymphatic vessel density in the surrounding stroma (PLVD) based on pT group. LVD—lymphatic vessel density; pT—pathological classification of the primary tumor size. Data are presented as mean ± SEM. *p* < 0.05 was considered statistically significant and is shown in red font.

**Figure 5 medicina-61-01712-f005:**
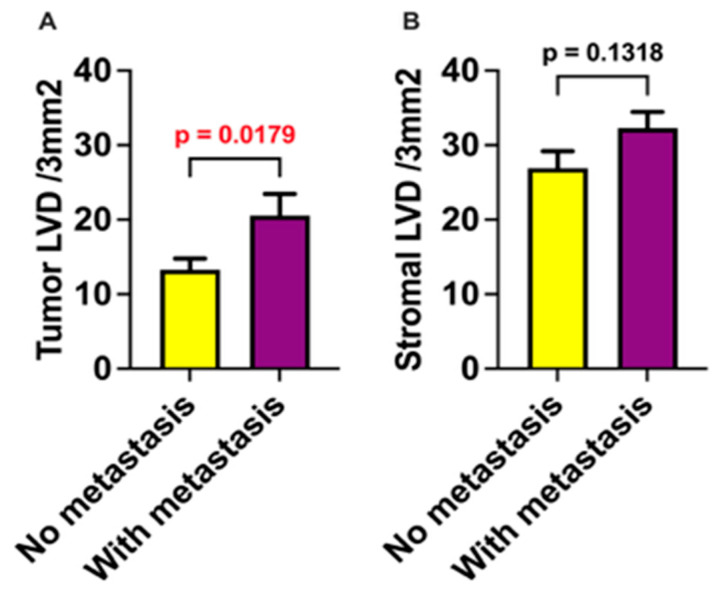
Intratumoral and stromal lymphatic vessel density (LVD) in relation to the presence of metastases. Patient tissue samples were stratified into two groups based on metastatic status: no metastasis (*n* = 28) and presence of metastasis (*n* = 13). (**A**) Lymphatic vessel density within tumor tissue (ILVD) in samples from patients without metastases. (**B**) Lymphatic vessel density in the surrounding stromal tissue (PLVD) in samples from patients with metastases. LVD—lymphatic vessel density. Data are presented as mean ± SEM. *p* < 0.05 was considered statistically significant and is highlighted in red font.

**Figure 6 medicina-61-01712-f006:**
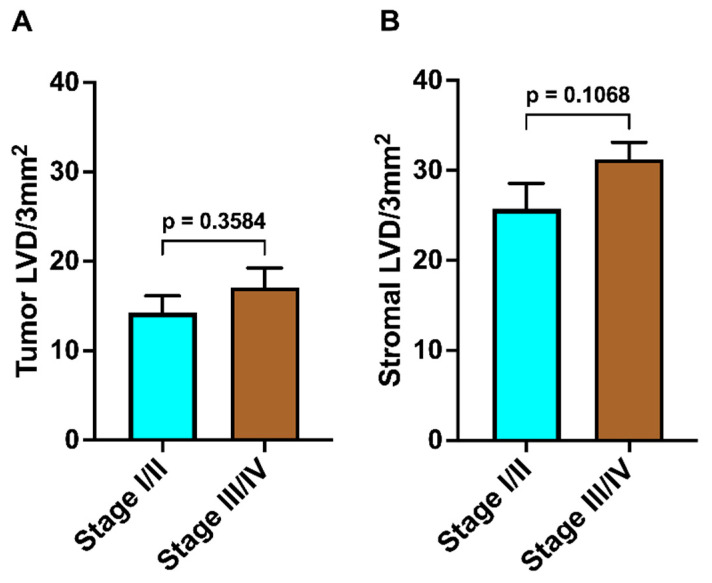
Intratumoral and stromal lymphatic vessel density in relation to the disease stage. Patients were stratified into two groups based on the stage of disease: stage I and II (low-stage, *n* = 19) and stage III and IV (high-stage, *n* = 24). (**A**) Lymphatic vessel density within tumor tissue (intratumoral LVD) in relation to overall disease stage (low-stage vs. high-stage) (**B**) Lymphatic vessel density in the surrounding stromal tissue (stromal LVD) in relation to overall disease stage (low-stage vs. high-stage). LVD—lymphatic vessel density. Data are presented as mean ± SEM.

**Figure 7 medicina-61-01712-f007:**
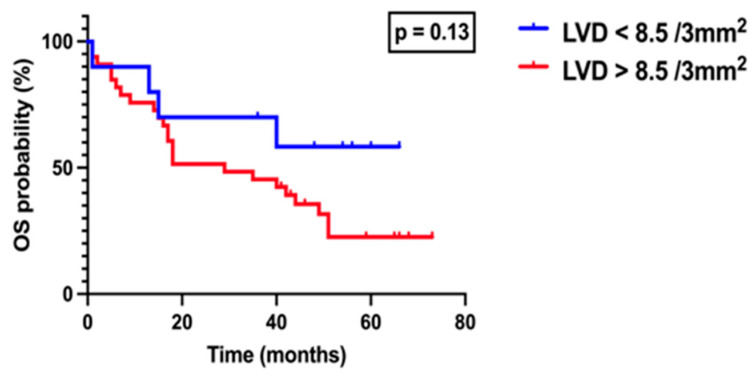
Kaplan–Meier overall survival curves stratified by intratumoral lymphatic vessel density (ILVD). Patients were divided into two groups according to tumor LVD based on the optimal cutoff value of 8.5 vessels/3 mm^2^, as determined by ROC analysis using the Youden index: LVD < 8.5 (blue line) and LVD > 8.5 (red line). Survival was analyzed using the log-rank (Mantel–Cox) test.

**Figure 8 medicina-61-01712-f008:**
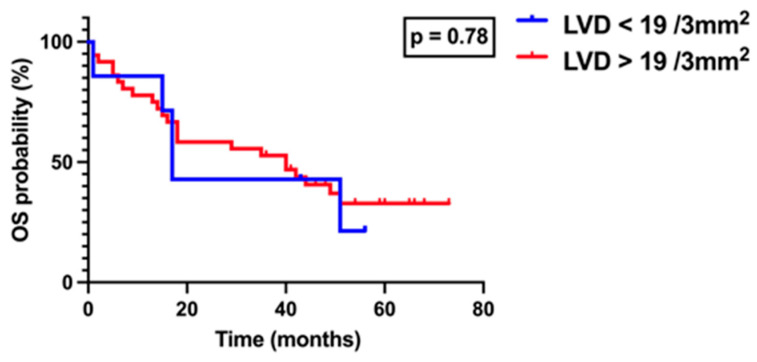
Kaplan–Meier overall survival curves stratified by peritumoral lymphatic vessel density (PLVD). Patients were divided into two groups according to tumor LVD based on the optimal cutoff value of 19 vessels/3 mm^2^, as determined by ROC analysis using the Youden index: LVD < 19 (blue line) and LVD > 19 (red line). Survival was analyzed using the log-rank (Mantel–Cox) test.

## Data Availability

The data that support findings of this study are available from the corresponding author upon reasonable request.

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
