# Peer review of "Prognostic Significance of Peritumoral and Intratumoral Lymphatic Vessels Density in Clinically Node-Negative (cN0) Oral Squamous Cell Carcinoma: A Preliminary Report"

_medicina, 2025, doi:10.3390/medicina61091712_

Round 1

Reviewer 1 Report

Comments and Suggestions for Authors

The authors Kos et al. report on a study with patients with oral cavity carcinoma (OSCC) and the occurrence of occult cervical metastases, as well as peritumoral and intratumoral lymphatic vessel density as a possible prognostic marker.

In my opinionn, this study is certainly interesting with regard to occult metastases, but it still presents several weaknesses.

  1. The authors refer to two possible treatment modalities regarding the neck, which is not correct. Recently, sentinel lymph node biopsy (SLNB)has also been applied in cT1 and cT2 OSCC. This should also be addressed accordingly.

  2. What were the inclusion and exclusion criteria? Clear definition please. How was a “reliable follow-up” defined?

  3. In the Methods section it is stated that in cT2 tumors, END was performed. However, the study also includes some pT1 tumors. Were these initially misclassified as cT2?

  4. Very heterogeneous group distribution with 21.0% (n=9) pT1 tumors, 48.8% (n=21) pT2, 9.3% (n=4) pT3, 18.6% (n=8) pT4a, and 2.3% (n=1) pT4b tumor. Were the final UICC stages compared?

  5. There are still some spelling errors, and the figures are incorrectly numbered (two times each for Figure 1 and 2).

  6. Were PLVD and ILVD compared between ENE+ and ENE– patients?

  7. The fact that 65.1% (n=28) of patients had lymph node metastases after initial cN0 Staging seems somewhat high, but that 9 of these also had extranodal extension is difficult to comprehend. This would mean that about 20% of the patients had “occult” metastases with ENE+ findings that were not recognized as such during staging. Were the CT/MRI images critically re-screened to check whether patients had been staged incorrectly?

  8. Were the subdivisions of patients and of PLVD/ILVD in the course of the analysis (see Figures 4 and 5) and their survival correlated again with stage?

Comments on the Quality of English Language

Proofreading is strongly advised, as there are still several spelling mistakes and inconsistencies in the figure numbering (Figure 1 and 2 exist twice)

Author Response

Thank you for reviewing our article entitled "Prognostic Significance of Peritumoral and Intratumoral Lymphatic Vessel Density in Clinically Node-Negative (cN0) Oral Squamous Cell Carcinoma: A Preliminary Report." We sincerely appreciate your valuable critique and have made efforts to address the points raised.

Comment 1:
“The authors refer to two possible treatment modalities regarding the neck, which is not correct. Recently, sentinel lymph node biopsy (SLNB) has also been applied in cT1 and cT2 OSCC. This should also be addressed accordingly.”

Response 1:
Thank you for this important comment. We acknowledge that we initially failed to discuss the use of SLNB adequately. Although SLNB is not currently used in our clinical practice, it is recognized as standard practice according to recent international guidelines. Therefore, we have now included a discussion of SLNB in the Introduction section, highlighting the available treatment options: elective neck dissection, SLNB, and watchful waiting, as you can see highlighted in red in the revised manuscript.

Comment 2:
“What were the inclusion and exclusion criteria? Clear definition please. How was a ‘reliable follow-up’ defined?”

Response 2:
Thank you for this valuable remark. We have clarified both the inclusion and exclusion criteria in the Materials and Methods section as follows:

  • Inclusion: (a) T1–T4 primary OSCC; (b) no previous history of Head and Neck Squamous Cell Carcinoma (HNSCC); (c) surgical resection of the primary tumor with concomitant END; (d) absence of neoadjuvant treatment; (e) no radiological or clinical evidence of LNM; and (f) reliable follow-up. 

  • Exclusion: (a) OSCC patients with a clinically node-positive neck (cN+); (b) previous surgical or chemo/radiotherapy of HNSCC; (c) incomplete data in hospital medical records. 

Reliable follow-up is now described in detail in Materials and Methods section, including scheduled clinical examinations every 2–3 months (first 2 years), then every 4–6 months (next 4 years), and annual thereafter, along with ultrasound and annual imaging (MSCT or MRI).

Comment 3:
“In the Methods section, it is stated that in cT2 tumors, END was performed. However, the study also includes some pT1 tumors. Were these initially misclassified as cT2?”

Response 3:
Thank you for this insightful question. It is indeed true that some tumors initially classified clinically as cT2 were found to be pT1 on pathological staging. However, discrepancies between clinical and pathological TNM staging are well documented in head and neck oncology (https://www.mdpi.com/2075-4418/13/13/2202). 

Comment 4:
“Very heterogeneous group distribution with 21.0% pT1 tumors, 48.8% pT2, etc. Were the final UICC stages compared?”

Response 4:
Thank you for this comment. We acknowledge the heterogeneity. To enhance clarity of our article, we further categorized patients according to the AJCC/UICC 8th edition overall stages of the disease (I–IV), now included and highlighted in the revised manuscript: 13.9% (stage I), 30.2% (stage II), 21.0% (stage III), and 34.9% (stage IV).

Comment 5:
“There are still some spelling errors, and the figures are incorrectly numbered (two times each for Figure 1 and 2).”

Response 5:
We appreciate your attention to detail and apologize for these oversights. The figure numbering errors have been corrected, and the manuscript has been thoroughly proofread to resolve spelling mistakes.

Comment 6:
“Were PLVD and ILVD compared between ENE+ and ENE– patients?”

Response 6:
Thank you for this valuable observation. We performed this analysis initially; however, the small number of ENE+ patients (n = 9) limited statistical power. As a preliminary study, we lack sufficient data to draw meaningful conclusions. We plan to include a larger cohort in future research to evaluate the association between LVD and regional disease aggressiveness, including ENE, number, and level of positive lymph nodes.

Comment 7:
“The fact that 65.1% of patients had lymph node metastases after initial cN0 staging seems somewhat high, but 9 also had extranodal extension, which is difficult to comprehend. Were CT/MRI images critically re-screened for staging accuracy?”

Response 7:
Thank you for this valuable comment. Upon review, we identified an error in the original manuscript: 65.1% refers to patients without lymph node metastases, not with. The correct figure for metastasis presence is 34.9%, with 60% of those exhibiting extranodal extension. We have corrected this and highlighted it in the revised limitations section, noting the small cohort size may influence distribution patterns. We did not manage to re-screen CT/MRI images for this study.

Comment 8: Were the subdivisions of patients and of PLVD/ILVD in the course of the analysis (see Figures 4 and 5) and their survival correlated again with stage?

Response 8:
Thank you for your valuable insight. In additional analysis, we compared both ILVD and PLVD between two groups of patients: those with low-stage (I and II) and those with high-stage disease (III and IV). We did not find a significant correlation between ILVD or PLVD and disease stage in these groups. We welcome any further suggestions you may have for addressing this issue. All changes are highlighted in red in the revised manuscript.

Reviewer 2 Report

Comments and Suggestions for Authors

Well presented and documented paper.

One thing that is missing is the number of lymphatic vessels in normal tissue of the oral region. Are there differences between the tongue, sublingual area etc? So it's important to evaluate the ILVD and the PLVD to the normal LVD and present/comment the findings.

 I have no other suggestion 

Author Response

Thank you for taking the time to review our article and provide thoughtful feedback. We sincerely appreciate your valuable input.

Comment 1:
One thing that is missing is the number of lymphatic vessels in normal tissue of the oral region. Are there differences between the tongue, sublingual area, etc.? So it's important to evaluate the ILVD and PLVD in relation to normal LVD and present/comment on the findings.

Response 1:
Thank you for highlighting this important aspect. We agree that comparing lymphatic vessel density in tumor tissue to normal oral mucosa would offer valuable insight into lymphangiogenesis in oral squamous cell carcinoma. As this is a pilot study, as noted in the title, we did not examine lymphatic vessel density in normal, healthy oral mucosa. However, we appreciate your suggestion and will include analysis of healthy stroma in our future research. Regarding differences among oral subsites (sublingual area, tongue, etc.), our current cohort is too small to reliably compare ILVD and PLVD across these locations. Nonetheless, we plan to address this question in a larger, future study as our patient series expands. Thank you again for your helpful recommendations.

Reviewer 3 Report

Comments and Suggestions for Authors

Dear authors,

I want to congratulate you for the scientific work. Unfortunately, oral cancer is a world wide disease that must be deeply understood and investigated.

The topic presented is very interesting, and the article brings a new perspective about the angiogenesis process in oral cancer. Your study is the few ones which demonstrate the peritumoral and intratumoral Lymphatic Vessels Density in Clinically Node-Negative (cN0) Oral Squamous Cell Carcinoma, being very original and relevant to the field. So i consider that the results can fill a gap present now in the scientific literature.

The introduction: -in order to be more specific and to bring a general idea about the histopathologic subtypes of oral cancer, on the introduction section, please discuss aspects regarding the oral cancer histopathological subtypes, as shown here DOI: 10.47162/RJME.61.4.22.

The citation of the references in text must respect the journal guidelines. 

The materials and methods section is very well structured. The figures must have a higher quality.

Why you did not find an association between DOI and ILVD, PLVD?

The results sub-sections must have a title(3.1, 3.2... are without titles).

The discussion section must start with the scope of the study.

The limitations of the study are clear stated.

The conclusions are consistent with the evidence and arguments presented
and they address the main question posed.

The references don t respect the journal template.

I hope that this study is only the beginning of other scientific work.

Author Response

Thank you for your kind comments and time you took to reveiw our article.

Comment 1: The topic presented is very interesting, and the article brings a new perspective about the angiogenesis process in oral cancer. Your study is among the few that demonstrate peritumoral and intratumoral lymphatic vessel density in clinically node-negative (cN0) oral squamous cell carcinoma, being very original and relevant to the field. I consider that the results can fill a current gap in the scientific literature.

Response 1: Thank you very much for your kind comments. We sincerely appreciate your positive feedback and recognition of our efforts.

Comment 2: In order to be more specific and to provide a general overview of the histopathologic subtypes of oral cancer, please discuss these aspects in the introduction, as shown in DOI: 10.47162/RJME.61.4.22.

Response 2: Thank you for your valuable remark. While the referenced paper covers a range of cancers located in oral cavity (including minor salivary gland carcinomas etc.), our study is focused exclusively on oral squamous cell carcinoma (OSCC), which is the most prevalent malignancy in oral cavity. Since further histopathological subtyping is not routinely reported in our pathology evaluations, and given our limited cohort size, this falls outside the scope of the present work. We have clarified this focus in the revised manuscript.

Comment 3: The citation of references in the text must follow the journal guidelines.

Response 3: Thank you for your observation. We have revised the reference citations to conform to the journal’s requirements.

Comment 4: The Materials and Methods section is very well structured. The figures must have higher quality.

Response 4: Thank you for this useful comment. We have regenerated all figures in GraphPad Prism at maximum resolution to improve quality.

Comment 5: Why did you not find an association between DOI and ILVD, PLVD?

Response 5: Thank you for this excellent question. Depth of invasion (DOI) was incorporated into OSCC staging with the 8th edition AJCC manual in 2017. However, our cohort (2010–2015) predates routine DOI reporting in our pathology records. As this is a pilot study, we were unable to analyze this parameter. We intend to include DOI in future, larger-scale research.

Comment 6: The results sub-sections must have titles (3.1, 3.2, etc. are without titles).

Response 6: Thank you for bringing this to our attention. We have now added appropriate titles to each results sub-section.

Comment 7: The discussion section must start with the scope of the study.

Response 7: Thank you for your suggestion. We have revised the Discussion to begin with a clear statement of the study’s scope, as highlighted in red in the revised manuscript.

Comment 8: The limitations of the study are clearly stated.

Response 8: Thank you for your kind comment.

Comment 9: The conclusions are consistent with the evidence and arguments presented and address the main question posed.

Response 9: Thank you again for your positive feedback.

Comment 10: The references do not conform to the journal template.

Response 10: Thank you for noting this detail. We have corrected all reference formatting to match the journal template.

Round 2

Reviewer 1 Report

Comments and Suggestions for Authors

Thank you very much for revising the article and for taking the recommendations/comments into account. Well done.

Reviewer 3 Report

Comments and Suggestions for Authors

The manuscript has been improved and can be published